# The Role of Trustworthiness Facets for Developing Social Media Applications: A Structured Literature Review

Angela Borchert *[ID] and Maritta Heisel

Department of Software Engineering, University of Duisburg-Essen, 47057 Duisburg, Germany; maritta.heisel@uni-duisburg-essen.de
* Correspondence: angela.borchert@uni-due.de

**Abstract:** This work reviews existing research about attributes, which are assessed by individuals to evaluate the trustworthiness of (i) software applications, (ii) organizations (e.g., service providers), and (iii) other individuals. As these parties are part of social media services, previous research has identified the need for users to assess their trustworthiness. Based on the trustworthiness assessment, users decide whether they want to interact with them and whether such interactions appear safe. The literature review encompasses 264 works from which so-called trustworthiness facets of 100 papers could be identified. In addition to an overview of trustworthiness facets, this work further introduces a guideline for software engineers on how to select appropriate trustworthiness facets during the analysis of the problem space for the development of specific social media applications. It is exemplified by the problem of "catfishing" in online dating.

**Keywords:** trustworthiness; trust modelling; computer-mediated introduction; social media applications; software development





## 1. Introduction

Trust plays a major role for the development of relationships [1]. It is considered a crucial factor when it comes to an individual's decision to initiate or continue an interaction with another party [2]. Trust is therefore particularly relevant in the context of social media, where human relationships are initiated or deepened [3]. This is especially true for those social media applications that introduce strangers online for interactions that might shift to the offline world. Such computer-mediated introductions (CMIs) have become increasingly popular in recent years and can be found in the business sector in the sharing economy (monetary exchange against goods or services provided by private people) or the private sector, for example in online dating (the search for amicable, romantic or sexual relationships) [4]. CMI platforms not only mediate the introduction of users, but also their trust-building and interactions [5].

Social media applications and CMI in particular hold a high number of risks for its users—not only because of the other users but also due to the service provider and the software application. In terms of other social media users, risks may involve psychological vulnerabilities like damaged self-esteem or a broken heart in online dating [6]. In addition, there are financial risks in terms of fraud, for instance when users do not receive a promised service in the sharing economy [6,7]. Performance risks describe for example services offered in the sharing economy that are not up to standard in a professional sense [7]. Moreover, health risks refer to sexually transmitted diseases, while physical risks involve robbery, sexual assault, violence, or rape when online-introduced people meet each other offline [6–9]. Concerning social media organizations that provide the service, users take privacy risks such as the potential malicious collection and use of personal information for additional economic gain [6,10,11]. Risks originating from the social media applications especially refer to security or information leakages depending on the applied technology [10,12].

For these reasons, assessing the trustworthiness of other social media users, the application itself, and the service provider is not only crucial to the decision to interact but also contributes to user safety [3]. It is assumed that the better an individual is able to assess the trustworthiness of counterparties, the better the decision basis for entering or continuing an interaction, so that associated risks can be reduced [13]. Trustworthiness assessments take place by evaluating relevant attributes of the counterpart for their extent of existence [5,14]. Trust-related attributes that can be linked to at least one of the parties mentioned above are known as trustworthiness facets [5]. If a user believes that a party possesses suitable facets in order to perform as desired and expected, the party is perceived as trustworthy and the foundation stone for trust is laid [14]. A trustworthiness assessment is generally a challenge. It is a subjective evaluation of ambiguous cues with no guarantee about its actual correctness [15]. It becomes even more complex when it can only be conducted via a software application. Cues for trustworthiness facets are different from what people are used to in the offline world [15]. Moreover, they are prone to manipulation, as social media users often try to present themselves in a better light online [16]. For these reasons, it is the responsibility of the software engineer to consider trustworthiness facets in the design of social media applications. Thereby, applications can support users in conducting a trustworthiness assessment and to overcome its challenges in the best possible way. However, as far as we know, software engineers lack an overview of trustworthiness facets for the software development process. Which trustworthiness facets are relevant for social media users to assess the trustworthiness of (i) other users, (ii) the social media application and (iii) the service provider? This knowledge is necessary for software engineers to develop graphical user interface elements or software features with whose help users can assess in specific scenarios when to trust another party [5]. Thereby, they can meet the users' need to perform trustworthiness assessments [17]. Still, even if such an overview exists, software engineers need guidance on how to select appropriate trustworthiness facets for specific scenarios. On these grounds, the objective of this work is to conduct a structured literature review to provide an overview of trustworthiness facets that can be related to individuals, organizations and software. In addition, this work provides a guideline for software engineers on how to select appropriate facets for incorporating or addressing them within software.

This work is structured as follows. In Section 2, we define trustworthiness facets and relate them to former trust research. Moreover, we point out related work on how users' trust development has been supported within software recently. Section 3 presents a literature review of the trustworthiness facets. Afterwards, Section 4 provides an instruction for facet selection for requirements engineering. In Section 5, we apply the guideline to identify relevant trustworthiness facets to the example of "catfishing" in online dating. Finally, we discuss this work in Section 6 and end with a conclusion in Section 7.

## 2. Theoretical Background

In this section, we first define trustworthiness facets. Afterwards, we relate the facets to trust in general and different types of trust in particular. Then, we refer to former approaches to how software supports its users in their trust development process.

### 2.1. Trustworthiness Facets

"Trustworthiness facets" is a term that is coined in the CMI context. They encompass attributes possessed by the involved parties of CMI, which are (i) organizations that present themselves on the CMI platform, like advertisers or the CMI service provider, (ii) the corresponding CMI software and (iii) CMI users [5]. Trustworthiness facets are desirable characteristics from which an individual can infer whether the parties involved are able and willing to act as desired and expected, and thus are trustworthy [5,14]. They may be personality traits or descriptive qualities. As CMI users assess the involved parties by their trustworthiness, the three parties take the role of so-called trustees while CMI users simultaneously are trustors, who may place their trust in the others [18].

Since the three trustees differ in their nature of existence as institution, technology, and individual, different trustworthiness facets have been identified by different streams of research. Social and organizational psychology, sociology, economics and computer science partly identified facets that can be led back to the factors of trustworthiness ability, benevolence and integrity [5]. The factors of trustworthiness are considered the attributes primarily associated with trustworthiness [5]. Originally determined for the interpersonal context, these attributes have been widely used for other trustee types, such as organizations and technologies [19–21]. Most often, researchers have adapted their definitions to their respective context and kept the terminology or renamed them for their purposes, such as competence for ability [22] or fairness for integrity [23]. As another example, Caldwell and Clapham determine competence, quality assurance and financial balance for organizational trustworthiness as a comparison to ability for interpersonal trustworthiness [22]. By these adoptions, some facets may have either the same terminology but different definitions or different terminologies despite describing a similar phenomenon.

Furthermore, former research identified facets for the various trustee types that are not based on the factors of trustworthiness. As an example, for trustworthiness facets of technology, Mohammadi et al. related software qualities to the trustworthiness of software [24]. Software qualities describe characteristics that enhance software [25]. They encompass characteristics necessary for the functioning of a system concerning the back-end as well as those for the front-end that are partly directly perceivable for the user [25].

### 2.2. Placing Trustworthiness Facets in the Context of Trust

Trustworthiness facets are in line with the trust definition of McKnight et al. [14]. They define trust as a reflection of an individual's beliefs about a trustee's possession of suitable attributes necessary to perform as expected in a given situation [5,14]. This reflection of beliefs is irrespective of the trustor's ability to monitor or control the trustee [5]. It rather depends on the trustor's personal characteristics and how she subjectively perceives the trustee's trustworthiness and related facets. This process of trustworthiness assessment is usually a bilateral exchange of trustor and trustee in terms of interpersonal interactions to result in an effective engagement. The trustee is interested in showing her trustworthiness by presenting her facets while the trustor perceives what the trustee reveals [26]. As trustor and trustee take simultaneously both roles in an interaction, it is a mutual process [27].

A trustworthiness assessment takes place when initiating or continuing an interaction [28]. It involves the evaluation of facets and the extent to which they are available. An unmet trustworthiness assessment may mean that trustworthiness facets are insufficiently available or irrelevant for the given situation. This may result in the impression of the party being untrustworthy and might lead to a termination of an interaction. Furthermore, the trustor may assess other attributes besides the trustworthiness facets, such as general personality traits, values, or goals. These are significant for the development of identification-based trust, which means that trust is established due to the fact that an individual identifies with a trustee [29]. If the trustor does not identify with the trustee, this might lead to the termination of an interaction, as well [29].

The process of evaluating trustworthiness facets differs in terms of timing. At the beginning of the first interaction, the trustor has no experience with the trustee. The trustworthiness assessment cannot draw on a knowledge base. Therefore, the interaction is foremost based on the trustor's first judgement of given cues by the trustee from which trustworthiness facets or other attributes can be inferred. These are relevant for cognitive categorization processes concerning the trustee on which basis the trustor develops initial trust [19,30]. Categorization processes can include (i) reputation categorization, (ii) stereotyping and (iii) unit grouping [19]. Reputation categorization makes use of second-hand information on which basis the trustor attributes trustworthiness facets to the trustee. Stereotyping means to put a trustee into a general category, which is also associated with certain facets. Unit grouping describes that the trustor places the trustee in the same category as herself. This positively affects the trustworthiness assessment, since the trustor

beliefs in sharing a social identity and, thus similar appreciated attributes and facets [31]. It can be concluded that although categorization processes and the trustworthiness assessment mutually define each other, perceived trustworthiness facets are strongly biased by cognitive processes.

In addition to categorization processes, calculative processes may also play a role leading to calculus-based trust [32]. The decision to start trusting and interacting with another party depends on rationally derived costs and benefits that may accompany or result from an interaction. The derivation process can be related to the trustee's predictability and further facets that influence the trustor's expectations concerning an interaction [32]. If the benefits exceed the costs, the trustor is likelier to interact with the trustee and extend her trust [33]. Again, a mutual influence can be observed between calculation processes and the trustworthiness assessment.

In general, it can be argued that initial trust is based on an insufficient knowledge base and assumptions made by the trustor [14]. Knowledge about the actual and past performance of the trustee are yet missing. This changes during an interaction when initial trust develops to knowledge-based trust. The knowledge base about the trustee increases and enables insights about potentially existent trustworthiness facets. On these grounds, the trustworthiness assessment is more reliable during the knowledge-based trust phase than during the initial trust phase [20]. Therefore, the facet-oriented approach can be particularly assigned to knowledge-based trust [14]. At that point, a trustor knows a trustee well enough to predict the trustee's behavior to a certain extent regarding a specific situation [30]. As trustor and trustee have a common history, knowledge-based trust is more persistent than initial trust. While initial trust strongly depends on rapidly changing costs and benefits and made impressions, knowledge-based trust is more stable when it comes to performance lapses or circumstance changes [29].

In the context of CMI, both initial trust and knowledge-based trust are relevant as people get to know and interact with each other on the CMI platform. For example, in online dating, the selection of a person of interest is rather driven by assumptions from the first impression, where profile photos or profile information serve as cues [34]. These are used for categorization processes, as for example gender stereotyping [35]. After first interactions, online dating users gain more knowledge about trustees during interactions, which facilitates the trustworthiness assessment. Some online dating platforms, such as affiny.co.uk or neu.de, allow their users to stepwise disclose more personal information, such as photos, when people feel more secure after having built knowledge-based trust to the other end-user first.

Initial and knowledge-based trust is also relevant for the interaction with CMI organizations and applications. Initial trust in a service provider is most often based on customer familiarity with the organization, its reputation, quality of provided information about the company and the service, certifications from third parties and attractive rewards for using the technology [36]. Trust in the service provider can foster initial trust in the software, which impacts why a user chooses to use a particular application [37]. During software usage, knowledge-based trust in both the service provider and the application can develop. According to Siau and Shen [36], trust in the service provider further develops while using its application by assessing the quality of the software, the provider's competence and integrity, appropriate privacy policies and security controls, the possibility of open communication and community building as well as external auditing. Trust in the software especially develops when it is perceived as reliable. Even if the trustworthiness assessment is not satisfactory concerning the service provider during software use, but for the software itself, the user may continue to use the application [14].

### 2.3. Trust-Building through Software in the Past

The development of trust between individuals through software has been researched and considered for software engineering in the past. Jones and Marsh introduced the TRUST notation for social media of Computer Supported Cooperative Work [38]. The

notation respects the elements knowledge, importance of interaction, utility of cooperation, basic trust and conceptual trust as a basis to record and evaluate interpersonal and group activities online. Thereby, TRUST serves as a tool for discussions in terms of software design for human-computer-human interactions.

In the context of e-commerce, Tran proposes a framework for trust modelling to protect buying agents from dishonest selling agents [39]. Their proposed algorithm is based on a trustworthiness threshold, which weighs among others price, quality and expected value of a product, as well as cooperation and penalty factor of an interaction. Thereby, the approach of Tran rather follows the principle of calculative-based trust. With their framework, buying agents may evaluate the trustworthiness of selling agents by software features including trust ratings.

Another trust modelling approach is the method for trust-related software features, called TrustSoFt, which is especially created for CMI [13]. Based on users' trust concerns, software goals are derived in order to oppose the user concerns. In addition, trustworthiness facets are identified that would counteract the concerns if they were present in the respective context. On these grounds, requirements and features for the software to be developed are elicited. Resulting software features shall support users in assessing the trustworthiness of involved parties as a strategy to cope with their concerns. The overview of trustworthiness facets presented in this work (see Section 3) can be regarded as a useful tool for applying TrustSoFt.

### 3. Literature Review of Trustworthiness Facets

With a structured literature review, we relate to the research question of what trustworthiness facets are evaluated by social media users to apply a trustworthiness assessment on other users, organizations and technology. In this section, we explain our procedure for the literature review and present our findings. For that, we follow the "Preferred reporting items for systematic review and meta-analysis" (PRISMA-P), whose checklist facilitates the report of a systematic review [40,41]. Figure 1 shows the PRISMA flow diagram for retracing the literature retrieval.

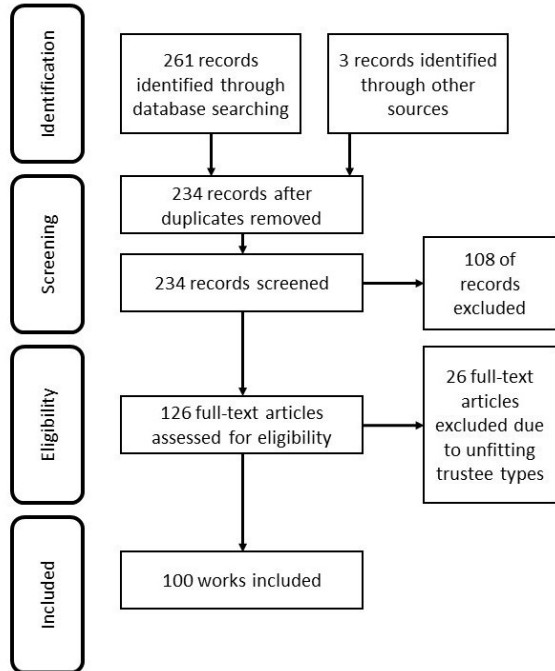

**Figure 1.** PRISMA flow diagram.

*3.1. Methodology*

For conducting the literature review, we oriented ourselves to the literature review guideline of vom Brocke et al., because it is especially suitable for research topics that comprise various disciplines with different backgrounds—such as both social and technical science [42].

We took one month for literature search and four months for the literature review. In terms of the eligibility criteria, we considered works that are published in research journals or conferences, written in English and independent of its publication year. As information sources, we used the databases Scopus and Web of Science. Furthermore, we especially respected the Journal of Trust Research and the Journal for Cyberpsychology, Behavior and Social Networking.

Our search strategy was to start with a keyword search. Keywords were categorized into trust-related (e.g., trustworthiness characteristics, trust determinants), individuals (e.g., social media users, interpersonal trust), technology (e.g., website characteristics, software qualities), and organizations (e.g., organizational trust, service provider). The keywords were combined with various operators (e.g., ("trustworthiness trait*" OR "trustworthiness characteristic*") and "individual"). Since the objective was to find trustworthiness facets for all three trustee types, trust-related keywords were combined with either one of the keywords for individuals, technology, or organizations. As soon as the keyword search led to an increasing number of same results, the next keyword combination was applied. In total, we had 13 rounds of keyword search. The keyword search was followed by a back- and forward search. Reference lists were screened for promising research titles concerning trustworthiness facets and works were considered that have cited prominent literature reviews that we found beforehand.

For study selection, we chose not only trust research about CMI or social media, which is partly in a preliminary state, but research about trustee characteristics in general. We accepted all application fields for our literature review as long as attributes could be related to trustees that are (i) organizations, institutions, companies, service providers or online vendors, (ii) software, applications, platforms, web sites or technologies or (iii) individuals in various roles. Regarding the latter, it must be mentioned that some works have analyzed trust in organizations by examining the relationship of employees and managers. We considered facets from such a research constellation as relevant for interpersonal relationships, if they are attributable to interpersonal interactions. For the selection, we first considered works whose titles included our trust-related keywords from keyword search or semantically referred to trustworthiness facets. Afterwards, we selected research that includes or describes trust-related attributes regarding one of the three trustee types in the context of (perceived) trustworthiness or trust development.

As a next step, the data collection process started. We scanned the selected literature for characteristics that were directly named or described as positively impacting (perceived) trustworthiness or as fostering trust development. Based on the trustee type discussed in the literature, we have documented the facets in lists for the corresponding trustee type. If available, we further documented the facet's definition, related other facets, the application field, used methods of the respective research and its reference. Some facets were documented multiple times depending on the frequency literature has mentioned them. We pre-defined trustworthiness facets as data items when literature has linked them to (perceived) trustworthiness or trust development by theoretical derivations, interviews or statistical calculations, such as correlations with a trust construct. For trustworthiness facets of technology, we only collected characteristics correspond to situational normality— i.e., software that meets certain quality standards that are considered customary by users for general success [14]. For us, this standard describes software that functions without failure on a technical side. On these grounds, we do not consider functional attributes for the static structure of a system that decrease its trustworthiness if not given. An example is scalability, which describes an increase in computing capacity to handle more operation in

a given time [14]. Instead, we narrow the overview of technological trustworthiness facets to those that are observable during operation.

Concerning the outcomes and prioritization, we were especially interested in literature that examined other trustworthiness facets than the original factors of trustworthiness ability, benevolence and integrity.

### 3.2. Results

Our study selection resulted in 264 works that have been published between 1957 and 2021. As a next step, we excluded literature based on either missing or imprecisely defined trustworthiness facets or characteristics related to trustee types or contexts not applicable for the here-described social media context (e.g., inter-organizational structures). In the end, we considered 100 works for the literature review. In terms of study characteristics, some of the selected works are also literature reviews, such as for example Fulmer and Gelfand [43], Mohammadi et al. [24], McKnight et al. [14], or Beldad et al. [44]. These reviews supported ours especially during the search phase. Other selected studies can be characterised as qualitative or quantitative research, while again others are theoretical discussions.

Our literature review results in three tables containing trustworthiness facets for each trustee type. They are presented in the Appendix A. Table A1 presents the trustworthiness facets for individuals, Table A2 for technology and Table A3 for organizations. Due to space constraints, we grouped trustworthiness facets representing similar phenomena based on semantically similar definitions. In literature, they are partly described as comparable to each other or as part of each other. In some cases, facets can be assigned to several groups (e.g., fairness). For each facet group, we formulated a universal definition that captures a group's entire essence. The definitions are to be understood as subjectively perceivable by individuals to a certain extent. It must be remarked that for the actual usage of single trustworthiness facets in software engineering, the precise definition from the original references should be considered. For each facet group, the first facet listed is the one that has been discussed most often of its group in research. The order of the other facets is random. The overviews include the grouped trustworthiness facets, their general definition, and references. We found a high number of references for the facets. For reasons of space, we limited the references to prominent works to which many of the non-referenced ones are traceable. We additionally included references that cover various research contexts.

For the overview of interpersonal facets (Figure 2), the research contexts of the reviewed literature are mainly related to certain roles such as trust in physicians, sales persons, managers or researchers. Research areas are for example communication, recruitment or initial trust. Trustworthiness facets of technology (Table A2) often stem from e-commerce, e-banking or social media. In terms of organizational trustworthiness facets (Table A3), many of them were analyzed in the context of online vendors, job application or employee perception.

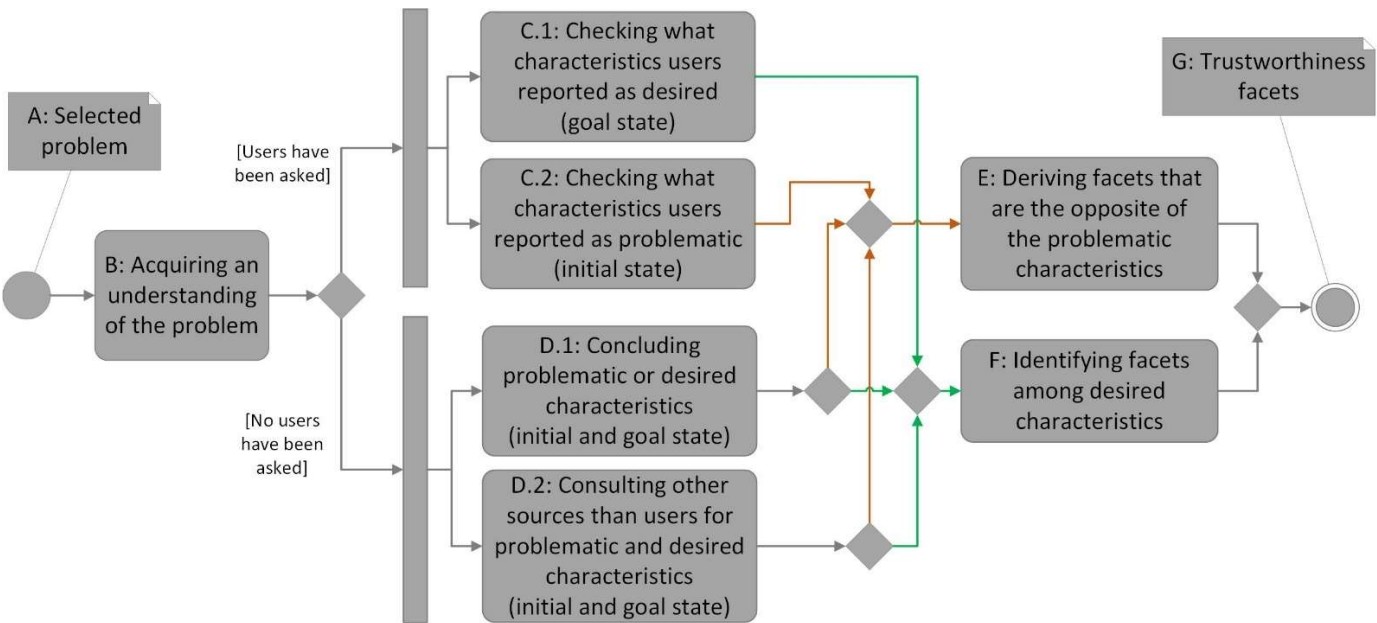

**Figure 2.** Guideline for identifying trustworthiness facets. Orange arrows point out the identification of problematic characteristics. Green arrows represent the paths for the identification of desired characteristics.

## 4. Guideline for Selecting Trustworthiness Facets for CMI Applications

In this section, we describe how software engineers can select appropriate trustworthiness facets to address or reflect them by software. Thereby, software can be designed in a way that its users are supported in their trustworthiness assessment of other parties at the platform.

The guideline is established based on the concepts of design thinking. Design thinking describes creative procedures for developing solutions concerning social needs [45]. It is a method that is regarded as valuable for IT development [46]. Among others, design thinking involves addressing human or environmental concerns, meaning that human needs should be targeted while situational factors are considered [45]. The first step is to clearly identify the problem space. This is regarded as a precondition to properly analyze the solution space for identifying solution approaches to a problem.

In order to understand a problem, an approach is to consider the initial state and goal state of a problem within the specific context [47]. The initial state describes the condition in which a conflict exists due to a discrepancy between two parties. In terms of a trust conflict, it occurs because of a mismatch regarding the trustee's characteristics (e.g., traits, attitude, values) or behavior and the trustor's expectations or preference [1]. The initial state is characterized by problematic characteristics causing the problem. Problematic characteristics can also be insufficiently available trustworthiness facets. In contrast, the goal state describes a condition in which a conflict between parties does not exist. The trustee matches the trustor's trustworthiness expectations and preferences. The goal state is characterized by traits that mitigate a concern to a point that it seems irrelevant. Such traits can be attributed to the context or trustor as well as to the trustee, i.e., they can also be trustworthiness facets. Initial state and goal state can be considered as opposite poles.

Design thinking is a valuable method for the identification of appropriate trustworthiness facets insofar that the facets are highly context-dependent. Hence, an extensive knowledge about the specific situation is necessary. This is covered by the analysis of the problem space, which is considered by the guideline. Figure 2 on page 8 presents the procedure of the guideline for software engineers by a UML activity diagram [48], which is explained in the following. For reasons of space and to maintain the clarity of the diagram, we used colored arrows. Orange arrows are used for explaining the identification

of problematic characteristics underlying the initial state, while green arrows point out the identification of desired characteristics that underlie the goal state.

### 4.1. The Problem

As a starting point, the software engineer must decide what problem the software to develop should address. In terms of the trustworthiness facets, this can be done by determining a trust conflict or a trust concern of users [13] (Box A in Figure 2). Regarding trust concerns, they describe an individual's specific expression of perceived risks that they evaluate as probable or critical in a particular application field or situation. These concerns exist, because the subject doubts the trustworthiness of the trustee [49,50].

### 4.2. Acquiring an Understanding

After the requirements engineer has decided on a problem they want to address in the software, they need to acquire an understanding of it (Box B in Figure 2). For that purpose, it is necessary to take over the user perspective as the problem shall be avoided in the future by the users' trustworthiness assessment. In requirements engineering, common techniques for obtaining an understanding of the context are user interviews, user surveys, analyzing former research, observing the application environment or creating user scenarios [51].

### 4.3. Identifying Facets Based on User Statements

If the software engineer decides to directly ask users for their perspective on the problem by interviews or surveys (upper part of activity diagram), we recommend to include questions about the initial state, goal state and their inherent traits, particularly trustworthiness facets. After the questioning, it must be checked whether users have reported desired traits referring to the goal state (Box C1 in Figure 2) or have mentioned problematic traits of the initial state (Box C2). If desired characteristics can be determined, the software engineer may identify trustworthiness facets among them or relate them to trustworthiness facets from the overview provided in Section 3 (Box F). In case that problematic traits can be specified, trustworthiness facets can be derived from them. For this purpose, those facets are to be considered whose definition on a semantic level represents the opposite of the respective problematic trait (Box E). The overview of trustworthiness facets from Section 3 serves as a tool for this. Both desired and problematic traits can result in trustworthiness facets (Box G).

### 4.4. Identifying Facets Based on Other Sources

If it is not within the scope of options to ask users about the problem, other resources must be considered for identifying related problematic or desired characteristics (lower part of activity diagram). Resources might be the engineer's own understanding of the problem (Box D1 in Figure 2) or other sources such as former research (Box D2).

With respect to the engineer's own previously acquired understanding (Box B), they must attempt to draw conclusions about problematic characteristics that cause the problem or about desirable characteristics that would make the problem nonexistent (Box D1). Problematic characteristics serve as a base for deriving trustworthiness facets by means of the overview from Section 3 as explained above (Box E). Desired characteristics are either trustworthiness facets or characteristics that refer to facets by either being similar to them or necessary for realizing them (Box F).

In addition to the engineer's own understanding, other sources might provide indications about relevant information of the initial or goal state of the problem (Box D2). These can either lead directly to desired or problematic traits or serve as a basis for their inference. If the other sources result in problematic characteristics, the requirements engineer can again check on trustworthiness facets with an opposite character (Box E). If the other sources lead to desired characteristics, it is again about checking whether trustworthiness facets are among them or can be related to them by using the overview of Section 3 (Box F).

Both approaches can lead to trustworthiness facets (Box G). Nonetheless, we recommend performing both approaches. On the one hand, own conclusions might complement other sources, which might not have directly addressed the problem in regard to the facets. On the other hand, additional input is likely to lead to a higher variety of results.

## 5. Example: Applying the Guideline and the Overview to Catfishing

We demonstrate the application of the guideline for selecting appropriate trustworthiness facets for the example catfishing in the context of online dating. In doing so, we relate to the steps of the guideline by referring to the boxes of Figure 2.

Catfishing is a suitable problem that we chose as an example because it has been identified as a trust concern of online dating users by former research (Box A) [52]. Catfishing refers to social media users, also known as catfish, who create a personal profile in which they pretend to have another identity for fraudulent or deceptive purposes [53]. The identification of trustworthiness facets that oppose catfish is relevant for designing CMI software that reduces the risk of catfish interactions. By picking up appropriate trustworthiness facets in software design, users can be supported in evaluating whether they are interacting with a catfish.

In order to obtain an understanding of catfishing (Box B), we analyzed former research. Simmons and Lee conducted an interview study with catfish and catfish victims [53]. They found that the catfish's motives are boredom, money fraud, or covertly checking up on others. Catfish victims often do not realize the scam until it is too late, such as when they meet offline or have a financial loss. Some sensed the deceit when the catfish was avoiding revealing her true identity, for example by a high number of excuses for video chats. As a result, the victims stopped the online interaction.

After acquiring an understanding of catfishing, we must decide whether we ask users directly for problematic or desirable characteristics stemming from catfish or the associated context or not. As it is out of our scope for this work to conduct a user study and Simmons and Lee have not reported any such characteristics in their interview study [53], we continue with the lower path of Table A3, where no users have been asked.

In order to derive characteristics based on our own understanding (Box D1), we reflect on the catfish definition mentioned before. Catfish are not what the online dater supposes them to be. As catfish act different to users' expectations at some point, catfish are unpredictable. Drawing this conclusion, unpredictability is a problematic trait of the initial state contributing to the catfish problem (Box D1). Having a look at the trustworthiness facet overviews in the Appendix A, we specify predictability as the facet that is the opposite of unpredictability (Box E). If an online dating app helps its users to assess whether another online dater is predictable and behaves as expected, the likelihood of falling for a catfish potentially decreases.

In addition to our own understanding, we looked for other sources about catfishing in online dating (Box D2). According to Schulman, who hosted the MTV television show "Catfish" from which the term originates, a catfish is characterized by dishonesty [54]. However, if the catfish decides throughout an online interaction to reintroduce honesty, the associated interpersonal relationship can recover [54]. On these grounds, we identify dishonesty as a problematic characteristic, while honesty is desired. Checking on the trustworthiness facet overviews in the Appendix A, honesty has been related to trustworthiness by former research (Box F). Successfully assessing another person's honesty would make catfishing less probable and may result in a growing relationship.

## 6. Discussion

This work addresses trustworthiness facets as an approach to analyze the problem space for developing safer social media software. Since trustworthiness facets are evaluated in people's trustworthiness assessment about others as a basis for interaction, knowledge about the facets is crucial for creating supportive social media. By reflecting trustworthiness facets via software, users are able to evaluate interaction partners and possible

consequences. Thereby, the software goal is to mitigate usage risks. Users, who are able to better assess the trustworthiness of other parties, are assumed to have a higher rate of desired and safe interactions with these parties.

In the following, we discuss trustworthiness facets in the background of the reviewed literature and how they relate to initial and knowledge-based trust in regard to CMI applications. In addition, we reflect on the proposed guideline and the trustworthiness facet overview in terms of the catfishing use case. Moreover, the leitmotif of software engineers while making use of the trustworthiness facets is discussed. Afterwards, we address limitations and future research. There, we give an outlook on how the facets can contribute to the solution space by being presented through software features.

### 6.1. Trustworthiness Facets in Regard to the Reviewed Literature

Trustworthiness facets are a complex construct, because they encompass trust-related characteristics of the three trustees (i) individual, (ii) technology and (iii) organizations, which may offer the before-mentioned technology. For this reason, an overview of trustworthiness facets is especially relevant for the development of social media applications, where all three parties are involved. Yet, former research considers trust-related characteristics, most of which relate to only one of the trustees. An example are other literature reviews such as Fulmer and Gelfand [43], Mohammadi et al. [24], McKnight et al. [14], or Beldad et al. [44]. These works give indications concerning the temporal state of research for the single trustee types. Such research has contributed a lot to this collection of trustworthiness facets, which has allowed us to gain a comprehensive overview for all three trustee types. This overview, together with the guideline for selecting appropriate trustworthiness facets, can serve not only as a tool for the development of social media applications but also for future research examining the interdependence of trustworthiness facets among the different trustee types. Knowledge about such interdependencies is valuable to gain a better understanding of the impact of the single facets. This in turn supports the design of effective software features that may address trust towards multiple trustees. As an example, Roy et al. found out that usability impacts both trust in web pages and web retailers [55]. On these grounds, focusing on usability during software development has a high impact on the users' trust development.

### 6.2. Evaluating the Guideline Regarding the Catfishing Example

The guideline for selecting appropriate trustworthiness facets can be regarded as a structured analysis of the problem space in terms of trustworthiness facets. Concerning the catfishing use case, software engineers gained an understanding of what catfish are, trust issues about them, and what trustworthiness facets must be available to identify another user to be not a catfish. The guideline proposes to follow three approaches, which are to ask users directly about catfish and problematic and desired characteristics, to consult other sources such as literature or experts about catfish or to acquire an own understanding. User studies acquire resources, such as suitable users that are involved with the problem or time, which we did not have. Thereby, we lost a significant source to understand the problem from an involved perspective. However, consulting other sources led us to the trustworthiness facet honesty, while we derived unpredictability from our own understanding. Finding suitable experts or literature is time-consuming. Moreover, the availability and accessibility of other sources has a huge impact on the results [56]. Yet, we regard it as a reliable way for software engineers to identify trustworthiness facets. In contrast, relying on one's understanding requires fewer resources, but carries the difficulty to be an unstructured process that is highly impacted by the engineer's subjective perspective and limited knowledge-base or expertise [57]. We recommend to not rely only on this approach and to increase its efficacy by group discussions with other engineers.

### 6.3. Trustworthiness Facets in Regard to Initial Trust and Knowledge-Based Trust within CMI Applications

As pointed out in Section 2, trustworthiness facets are of high relevance for the development of initial and knowledge-based trust. For social media applications like CMI, this is important insofar as both kinds of trust occur online—especially in terms of interpersonal relationship development. Although initial trust is mainly based on mere assumptions, CMI users usually quickly transfer their interactions from the online platform to the physical world [58]—oftentimes before knowledge-based trust has evolved. The risk of unpleasant incidents is likely to be higher as when CMI users waited until they obtained more knowledge about the other users online.

On these grounds, we propose to use the knowledge about trustworthiness facets in two ways. First, the introduction phase between users can be enhanced. CMI software could focus more on asking for and providing information about facets of a user to provide a better understanding of the person. In addition, software could enable users to directly communicate their expectations and desired facets of other users for facilitating trust development and avoiding trust issues in the long run. This could be realized by displayed information in the user profile or proposed ice-breaking questions in the CMI chat. Another possible software feature is to describe scenarios and ask users how they would behave in such a situation. Answers can be displayed in the profile, allowing users to learn more about each other and making cognitive processes of initial trust more accurate. The two latter software features address the trustworthiness facets predictability and integrity. Users can better estimate whether a match fits to their conceptions, wishes and norms. In addition, such software features provide CMI users with initial knowledge about the other users even if they decide to directly have an offline encounter without building knowledge-based trust. If online dating users discover consistencies between online dating profiles and a users' offline behavior in a first face-to-face date, the facet honesty is proven, making the development of meaningful relationships more likely [59]. Nevertheless, there is a risk of deception and dishonesty being uncovered in the first face-to-face date, as in the catfishing example [59]. In this case, proving honesty offline might be too late. Software features for the authentication of user profiles have been identified as a counter measure that reduces user concerns about catfishing and refer to users' honesty [60]. Such features may include an algorithmic or manual comparison of uploaded profile pictures with photos created directly in the application. Positive authentication is confirmed with a tick icon in the user profile. This procedure is used by various online dating apps, such as Tinder (www.tinder.com).

Along with fostering initial trust building on the platform, it is equally important to encourage and assist users to develop knowledge-based trust online—prior to offline encounters. Indeed, a positive effect of longer lasting online interactions on the satisfaction regarding relationship building is observed by online dating users during the COVID-19 pandemic [60]. A way to realize this can be to activate software features for self-disclosure gradually during the online interaction when a certain level of trust has been already established. As an example, user photos are blurred for other users at the online dating platform Affinity (www.affiny.co.uk) until the respective user decides to reveal them. This allows users' curiosity for information to be satisfied safely and reduces risks such as identity theft through pictures. At the same time, user engagement with the CMI application can be encouraged. User engagement is desired by social media service providers as it is a significant driver in their business model [61]. Users contribute more data, are longer exposed to advertising of the service provider's partners and are more likely to make purchases on the platform.

In the context of the sharing economy platform Airbnb, Mao et al. found that it is worth it for online vendors to make effort in increasing users' trust in hosts (i.e., CMI users who rent accommodations to other CMI users) and trust in the platform [62]. Both positively impact users' repurchase intention after users have established a knowledge-base about hosts and the platform already, while trust in the service provider is detached from

that. Mao et al. identified calculus-based trust, experiences shared together, perceptions of reliability and dependability, and positive expectations due to personality traits as antecedents of trust in hosts. For trust in platform, they identified perceived security, perceived privacy, perceived web-site quality, and social presence as relevant. Some of these antecedents are or can be related to trustworthiness facets of the overview provided in this work. On these grounds, service providers can increase their financial gain by implementing software features that address these trustworthiness facets.

Altogether, supporting users' trustworthiness assessment within the CMI application is in the interest of both the CMI service provider and the CMI user.

### 6.4. Misuse of Trustworthiness Facets

As argued before, considering trustworthiness facets in software has many positive aspects. Users are able to better assess the parties involved, leading to safer CMI use, facilitated development of trust relationships, increased user engagement, and greater financial gain for service providers. Therefore, it is all the more appealing to exploit the knowledge about the trustworthiness facets in one's own interest. It could be misused to portray oneself as trustworthy, even though this might not be the case.

CMI users may make up their trustworthiness for reasons of fraud or making fun of other users as in the case of catfishing [9,63]. Service providers may set cues on the platform referring to trustworthiness facets such as benevolence or transparency to promote themselves. In doing so, they may increase their economic gain without actually caring about the users or being transparent as they pretend to be. Software engineers could misuse the trustworthiness facets to make an application appear trustworthy in terms of security or privacy on the front-end, even though the back-end is not aligned with it. Depending on the extent to which the presentation deviates from reality, ethical or legal violations may occur.

Dealing with such misuse is difficult, because we cannot control how the provided knowledge will be used. We want to emphasize that the knowledge should only be used to provide indications to the question of whether a trustee is truly trustworthy. Software features that pick up the trustworthiness facets should support users in their trustworthiness assessment neutrally. This means that the software may provide users the possibility for an honest self-presentation as far as possible or aid parties in truthfully demonstrating trustworthiness facets online. This can be realized by asking users to perform certain actions with which they can prove the possession of trustworthiness facets that can be testified by third parties. As an example, the dating website match.com (www.match.com) offers a personality test on scientific basis. After answering it, the level of various personality traits is depicted in the users' profiles. Other users have access to the results, which supports their trustworthiness assessment and the matching process. Another example is again authentication processes where users can verify their identity to the service provider by uploading their ID. In turn, they receive a graphical icon in their profile that demonstrates their integrity with the platform's standards and their honesty regarding their provided personal information.

Besides a truthful application of the trustworthiness facets, we propose that users should be made aware of the possibility of misrepresentation in self-reports within the application. Warning messages can be useful instruments for this purpose. Moreover, legal regulations could be helpful to reduce untruthful self-reporting. Future research needs to address how ethical violations can be detected and prevented from a technical perspective.

### 6.5. Limitation and Future Work

Although the overview of trustworthiness facets is a valuable asset for requirements engineering of social media applications, their practical usefulness has to be evaluated further. Trust research is a vast field. Nonetheless, research on the trustworthiness facets in social media, especially CMI, is still emerging. Since previous research has transferred trustworthiness facets from the original domain to another application field before, we

decided to also review facets that are not unique to social media or CMI. This has the advantage that we were able to identify a large set of facets. However, due to their strong context dependency, it has to be validated whether they are really applicable to the field of social media and CMI. In addition, the guideline for selecting adequate facets must be validated as well. Its practicability and efficiency of outcome needs further examination in future work.

Moreover, the overview may lead to the belief that the reviewed facets have equal relevance in the trustworthiness assessment or trust development. Yet, we collected trustworthiness facets in general without distinguishing between their level of impact regarding trust or trustworthiness. Most often, the literature being reviewed did not provide this knowledge. However, there is reason to suspect that some facets have a more decisive role in the trustworthiness assessment than others. Borchert et al. found that in software development, the choice of trust concerns and trustworthiness facets to be addressed by software varied the degree to which users trust an application [64].

In terms of future work, we plan to continue with trustworthiness facets that emerge from the problem space and use them as input to the solution space. The question is how the facets can be addressed or reflected within the software. One possible approach could be via software features. While in this work we proposed exemplary software features for addressing trustworthiness facets based on brainstorming and creativity, we are striving to develop a structured method of how exactly software features for the trustworthiness assessment in social media can be derived from trustworthiness facets.

## 7. Conclusions

Trustworthiness facets are considered essential cues for assessing the trustworthiness of parties with whom one interacts, such as other people, technologies, or organizations. It is assumed that the better a social media application supports its users in their trustworthiness assessment, the safer and more promising user interactions will be. This work provides an overview of trustworthiness facets based on a literature review. In addition, we introduce a guideline for software engineers for selecting appropriate facets for their application field. The overview and guideline can be regarded as tools for the analysis of the problem space in the beginning of the software engineering life cycle. In this process, a finding is that problematic characteristics causing user concerns can be used for specifying relevant trustworthiness facets in order to enhance user interactions. With our research, we are one step closer to developing social media applications that respect users' psychological processes during relationship building. In future work, we want to pick up trustworthiness facets as input for a structured analysis of the solution space during software development. We aim to systematically elicit software features that support users in their trustworthiness assessment via social media applications. As the literature review covers general research on trustworthiness facets as well, we can then assess its validity in the context of social media development.

**Author Contributions:** Conceptualization, A.B. and M.H.; methodology, A.B.; validation, A.B. and M.H.; formal analysis, A.B.; investigation, A.B.; data curation, A.B.; writing—original draft preparation, A.B.; writing—review and editing, M.H.; visualization, A.B.; supervision, M.H.; project administration, A.B.; funding acquisition, M.H. All authors have read and agreed to the published version of the manuscript.

**Funding:** This research was funded by Deutsche Forschungsgemeinschaft (DFG, German Research Foundation) grant number GRK 2167, Research Training Group "User-Centred Social Media". Moreover, we acknowledge support by the Open Access Publication Fund of the University of Duisburg-Essen.

**Institutional Review Board Statement:** Not applicable.

**Data Availability Statement:** Not applicable.

**Conflicts of Interest:** The authors declare no conflict of interest.

## Abbreviations

The following abbreviations are used in this manuscript:

| | |
|---|---|
| CMI | Computer-Mediated Introduction |
| TrustSoft | Method for Trust-Related Software Features |
| PRISMA-P | Preferred Reporting Items for Systematic Review and Meta-Analysis Protocols |
| UML | Unified Modeling Lanuage |

## Appendix A

The overviews of trustworthiness facets that resulted from the literature review are presented in following in Tables A1–A3.

**Table A1.** Overview of trustworthiness facets of individuals.

| Trustworthiness Facets | Definition | References |
|---|---|---|
| Ability, competence, expertise, knowledge, skill, wisdom, business sense, influence, power | Skills or characteristics that enable to fulfill obligations or to have impact in a specific domain. | [5,65–75] |
| Accessibility, approachability, attentiveness, availability, openness, receptivity | Being physically present when needed, mentally open and receptive, easy to talk to and a careful listener. | [20,26,68,69,71,74–76] |
| Attractiveness | Being appealing to others. | [20,73,77] |
| Benevolence, availability, candor, care, loyalty, openness, receptivity, agreeableness, selflessness, honesty, altruism, goodwill | Having concerns about others, wanting something good for others and acting in their interest without an egocentric motive. | [5,20,24,26,69,71,73–76,78,79] |
| Confidentiality, discreetness | Entrusted knowledge is kept in confidence. | [68,69,80] |
| Emotional stability | "[B]eing calm, enthusiastic, free from anxiety, depression and insecurity" [68] | [73] |
| Empathy | The ability to comprehend feelings of others. | [26] |
| Extraversion, dynamism | Talkativeness, sociability, friendliness | [73,74] |
| Honesty, credibility, truthfulness, authenticity, openness, accuracy, willingness to disclose | Correctness of information and freely sharing information and ideas | [20,26,66,68,70–72,76] |
| Humbleness | The notion to not take oneself more important than others | [27] |
| Integrity, fairness, consistency, reliability, discreetness, morality, ethicality, credibility, honesty | The trustee complies to the trustor's accepted principles (e.g., moral, ethical) that are predictable and reliable leading to equity. | [7,26,65,67–69,72–75,78,81] |
| Justice, fairness | The trustee morally respects the trustor's interests and the trustor herself—especially concerning provided information and interactions. | [68,82,83] |
| Likability, rapport | Friendliness, high sympathy and a person with whom the trustor wants to spend time together and cooperate. | [71,84,85] |
| Predictability, consistency, reliability, good judgement, promise fulfillment, dependability, conscientiousness, performance | A stability in one's actions that is based on recurring behaviour, the ability to make good decisions, being productive and carrying out responsibilities reliably. | [68,69,73–76,79,86] |
| Popularity, social desirability | Social or cultural approval, socially desirable | [74,79] |
| Reputation | The perceived identity of a trustee which reflects personality traits, behaviour or presented images that is based on the trustor's own observations over a period of time or on secondary sources. | [19,87] |
| Respectfulness | The trustee regards "others and their perspective as valuable" [26] | [26] |
| Similarity, shared understanding, share of values | Perception of shared interests, values, appearance, lifestyle, status or culture | [20,65,71,72,85,88] |

**Table A2.** Overview of the trustworthiness facets of technology.

| Trustworthiness Facets | Definition | References |
|---|---|---|
| Ability, competence, expertise, credibility | The system is believed to have the skills and expertise to perform and act effectively in specific domains and to fulfill its promised services and responsibilities. Based on that, the user accepts its advice and believes its output. | [14,21,56,57,85,89–92] |
| Benevolence, helpfulness, goodwill | The system acts in the user's interests, cares for him/her, is well-intentioned and provides help or guidance when needed. | [14,21,56,57,85,89] |
| Information quality, content quality, data-related quality (consists of data integrity, data reliability, data timeliness, data validity), usefulness | The system provides sufficient information that is accurate, understandable, useful, complete, relevant and timely updated so that the user is able to evaluate the context (e.g., product, service, seller). | [14,24,63,85,93,94] |
| Integrity, compliance, compatibility | The system complies with standards (e.g., industry specific standards) or regulations, adheres to the user's accepted ethical or moral codes and is compatible with his/her beliefs or values. | [14,21,24,56,57,77,90] |
| Non-Repudiation | Ability to prove to sender that data has been delivered and to prove to receiver the sender identity for an unambiguous data transmission. | [24] |
| Openness, transparency | The system provides how it works and complies to standards and regulations. | [24,93] |
| Performance, reliability, predictability, dependability, functionality, accuracy, availability, failure tolerance, accountability, responsiveness, result demonstrability, correctness | The system executes correctly to accomplish the service that it promises. It is predictable despite potential failures and delivers proper outputs. | [14,15,21,24,56,77,85,95–99] |
| Privacy, confidentiality | Privacy refers to the provision of information and the risk of its exposure to unintended parties. Systems, which respect their users' privacy, limit the access of the users' data to only authorized agents and enable users to take control of its usage. | [24,63,85,92,96,98,99] |
| Reputation, image, brand strength, visibility | On the one hand, the technology's recognition and how much it might enhance the user's social status. On the other hand, an "easy identification of the [associated] company and its activity sector" [98]. | [77,92,97,98,100] |
| Safety | The system operates in a way that keeps its users' life and property safe and does not risk any harm or injuries. | [15,24,63] |
| Security, confidentiality | The system knows its users' vulnerabilities and protects them and their resources against attacks, misuses and unauthorized access. | [15,24,63,85,92,95,96,99,101,102] |
| Situational normality, social presence | The perception that the system is "normal, proper, or suited to a successful venture" as well as "personal, sociable, and [has] sensitive human elements, creating a feeling of human touch" [58]. | [20,21,63] |
| Usability, comprehensibility, effectiveness, ease-of-use, efficiency | A system designed in a way that enables users to effortlessly use it with easy access to understandable information that supports users in the usage. | [21,24,57,92–94,96,97,102] |
| Website quality, completeness, perceived usefulness, web site design, interface design, likeability | On the one hand, the extent to which the implemented set of software features meet the needs of its users. One the other hand, an attractive graphical design in terms of structure, navigation and content. | [14,21,24,63,85,94,96,97,102] |

**Table A3.** Overview of the trustworthiness facets of organizations.

| Trustworthiness Facets | Definition | References |
|---|---|---|
| Ability, competence, financial balance, quality assurance | Knowledge and skills to provide the service or product promised by the organization (while being both effective and efficient in regard to expended costs) | [19,20,22,23,75,92] |
| Benevolence, concern, goodness, morality, caring, interactional courtesy, responsibility to inform | Respecting and showing respect to the interests of the consumers and not taking advantage of their vulnerability. | [19,20,22,75] |
| Familiarity, similarity | Perception of same values or interests | [66,78,85,103] |
| Integrity, (procedural) fairness, justice, legal compliance, structural assurance | The existence of principles, values, standards or regulations (e.g., law, organizational policies, organizational procedures, contracts) to which an organization corresponds as promised. This most often relates to a high quality of treatment and equity. | [19,20,22,23,68,81,83,95,104,105] |
| Openness, honesty, transparency, confidential information sharing, responsibility to inform, comprehensibility | The availability, simplicity or clarity of information disclosed by an organization that allow individuals to comprehend the performance or internal workings of that organization. | [19,22,71,74,75,106] |
| Performance | Current actions for providing a service or product, which may involve the delivery, relative costs and the performance of the service/product itself. | [71] |
| Reliability, credibility, consistency, dependability, responsibility, predictability | The organization complies by its actions with its promises and offers guidance and support in times of crisis. | [19,23,75,102,104] |
| Reputation, prototypical organizational identity, brand image | Perception of an organization's culture, attributes, beliefs, values or prestige based on customer's own experience or hearsay from secondary sources. | [20,71,74,78,83,93,96,97] |
| Responsiveness, interactivity | Being responsive to the customers' requests and providing rapid feedback | [22,85,93,98] |
| Security | The organization provides a comfortable, assured and safe feeling | [20] |
| Situational normality | The individual's belief of an organization's success based on the perception how customary a situation with the organization seems to be | [19,102] |
| Size | The larger a company overall size and its market share position, the more experience it seems to have leading to a higher perception of trustworthiness. | [71,96,97,102] |
| Willingness to customize, service customization | Specialized equipment or adaptation of production processes or services to meet the customer's needs. | [71,93,96–98,102] |

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
