# Peer review of "The Role of Trustworthiness Facets for Developing Social Media Applications: A Structured Literature Review"

_information, doi:10.3390/info13010034_

Round 1
Reviewer 1 Report
The paper: The Role of Trustworthiness Facets For Developing Social Media Applications, presents the results of a review research regarding the attributes used for evaluating the trustworthiness from individuals perspectives.
The topic is interesting, however there are some aspects that should be improved:
- there are not research questions/research hypotheses. Please add and organize the literature review section and the results based on the research questions.
- From my point of view the tables that are presenting different definitions could be in the appendix part.
- The research methodology is not clear. Which were the steps of the research. At this stage is difficult to replicate.
- Please emphasize the originality of the manuscript. What gap in the literature it fills.
Good luck!
Reviewer 2 Report
The topic is interesting and the structure of this paper is good. But I think some comments are listed need to revise.
1.The authors discuss about the role of trustworthiness facets for developing social media Applications. The author should increase the discussion and explanation for the research limited part of the case study.
2.In the literature review of trustworthiness facets, the author should be detailed literature discussions, and it would be better if the literature can be added in the last five years.
3.The writing of the conclusion can be improved. The conclusion should include following contents such as background, research objective, experiment result, finding and future research and limitations. Please confirm it.
4.Please confirm that the format of this article meets the specifications required by the journal.
5. To be legible, please edit your manuscript to prevent redundancies, grammatical errors and punctuation problems.
Reviewer 3 Report
In light of AI advances and the era of digitalization this paper highlights and highly important aspect of trust in computer-mediated interactions. The topic is highly relevant.
R1. A general comment: As this is a (scoping) review I would recommend the authors to follow PRISMA or a similar methodology. Moreover, the authors should work on the workflow of the paper. Section 3 is a mixture of methodology and background. Section 4 is a mixture of the results of the literature review process and the author's guidelines.
R2: As this is a literature review I would recommend the authors to follow PRISMA or a similar methodology. (''This work reviews existing research about attributes, which are assessed by individuals to evaluate the trustworthiness of I) software applications, ii) organizations (e.g., service providers)
and iii) other individuals. A'')
Regardless of that key information on the methodology of the research and paper section process, in particular, is missing
DESIGN: e.g. which scientific repositories were used, what were the search queries
CONDUCT: e.g. what was the selection process, how many rounds, and who performed the process
ANALYSIS: type of information required, methodology of the process
I would strongly advise the authors to at least review the relevant methodologies. ''Although review articles can be organized in various ways, some generalizations can be made. All authors are expected to follow accepted conventions for reporting on how the study was undertaken. It is necessary to describe transparently the process of designing the review and the method for collecting literature, that is, how the literature was identified, analyzed, synthesized, and reported by the author '' (Snyder, H. (2019). Literature review as a research methodology: An overview and guidelines. Journal of business research, 104, 333-339, page )
R3. Discussion should incorporate more of the research carried out in this work, i.e. what are the Trustworthiness Facets, How does the list differ from what was previously identified, etc. At the moment discussion only partially fits the title of the paper, the extensive work carried out nor the motivation expressed.
R4.The motivation is clearly expressed. However, I would revise the introduction. Please add relevant literature to claims such as:
''In terms of other social media users, risks may involve26
psychological vulnerabilities like damaged self-esteem or a broken heart in online dating.In addition, there are financial risks in terms of fraud, like when users do not receive a promised service in the sharing economy.''
''For these reasons assessing the trustworthiness of other social media users, the application itself and the service provider is not only crucial to the decision to interact but also contributes to user safety''.
''Since the application is developed by the service provider, supporting both user interaction and trustworthiness assessment to increase user benefit and safety can be seen as corporate social responsibility.''
R5. ''Since the application is developed by the service provider, supporting both user interaction and trustworthiness assessment to increase user benefit and safety can be seen as corporate social responsibility.'' The authors are assuming only applications and UIs developed by the social media providers are used. What about apps developed by the community?
R6. The paragraph on Page 2 line 54, does not fit well into the narrative. It reads just like a ''random'' statement placed in the introduction. Even more so, since it starts with ''however'' one would accept it would be related to the previous paragraph.
R7. The background is in general well highlighted, however, when it comes to the justification of research, the existing works are represented superficially. i.e. ''Especially other literature reviews gave us further indications for our search. These were for example Fulmer & Gelfand [40], Mohammadi et al., [41], McKnight et al., [42] or Beldad et al. [43]. ''. This fits into the background.
Round 2
Reviewer 1 Report
Dear author/s,
thank you for the improved version of the manuscript. Please pay attention to the format of the paper.
Good luck!
Author Response
Thank you very much for your review.
We checked on the format.
Reviewer 3 Report
I would like to thank the authors for updating the manuscript. In principle, I have no comments. I would just add minor remarks:
1) Please add a literature review to the title
2) A PRISMA Diagram would be welcome to visualize the review process
3) Before Section 6 The paragraph is bolded.
Author Response
Thank you very much for your review.
1. We adjusted the title to:
The Role of Trustworthiness Facets For Developing
Social Media Applications: A Structured Literature Review
2. We added a PRISMA flow diagram (Figure 1)
3. We checked on the whole format.